# Fecal Microbiota and Associated Volatile Organic Compounds Distinguishing No-Adenoma from High-Risk Colon Adenoma Adults

**DOI:** 10.3390/metabo13070819

**Published:** 2023-07-04

**Authors:** Kyriaki Katsaounou, Danae Yiannakou, Elpiniki Nikolaou, Cameron Brown, Paris Vogazianos, Aristos Aristodimou, Jianxiang Chi, Paul Costeas, Agapios Agapiou, Elisavet Frangou, George Tsiaoussis, George Potamitis, Athos Antoniades, Christos Shammas, Yiorgos Apidianakis

**Affiliations:** 1Department of Biological Sciences, University of Cyprus, Nicosia 2109, Cyprus; katsaounou.kyriaki@ucy.ac.cy; 2Stremble Ventures Ltd., Limassol 4042, Cyprus; danae.yiannakou@stremble.com (D.Y.); cameron.brown@stremble.com (C.B.); paris.vogazianos@stremble.com (P.V.); aristos.aristodimou@stremble.com (A.A.); athos.antoniades@stremble.com (A.A.); 3AVVA Pharmaceuticals Ltd., Limassol 4001, Cyprus; e.nikolaou@avvapharma.com (E.N.); c.shammas@avvapharma.com (C.S.); 4Karaiskakio Foundation, Nicosia 2032, Cyprus; jason.chi@karaiskakio.org.cy (J.C.); paul.costeas@karaiskakio.org.cy (P.C.); 5Department of Chemistry, University of Cyprus, Nicosia 2109, Cyprus; agapiou.agapios@ucy.ac.cy; 6Nicosia General Hospital, Nicosia 2029, Cyprus; elislida@cytanet.com.cy (E.F.); tsiaoussisgeorgios@gmail.com (G.T.); 7Nikis Center, Nicosia 1082, Cyprus; g.potamitis@logos.cy.net

**Keywords:** dysbacteriosis, pathobionts, nutrients, metabolites

## Abstract

Microbiota and the metabolites they produce within the large intestine interact with the host epithelia under the influence of a range of host-derived metabolic, immune, and homeostatic factors. This complex host–microbe interaction affects intestinal tumorigenesis, but established microbial or metabolite profiles predicting colorectal cancer (CRC) risk are missing. Here, we aimed to identify fecal bacteria, volatile organic compounds (VOC), and their associations that distinguish healthy (non-adenoma, NA) from CRC prone (high-risk adenoma, HRA) individuals. Analyzing fecal samples obtained from 117 participants ≥15 days past routine colonoscopy, we highlight the higher abundance of Proteobacteria and *Parabacteroides distasonis*, and the lower abundance of *Lachnospiraceae* species, *Roseburia faecis*, *Blautia luti*, *Fusicatenibacter saccharivorans*, *Eubacterium rectale*, and *Phascolarctobacterium faecium* in the samples of HRA individuals. Volatolomic analysis of samples from 28 participants revealed a higher concentration of five compounds in the feces of HRA individuals, isobutyric acid, methyl butyrate, methyl propionate, 2-hexanone, and 2-pentanone. We used binomial logistic regression modeling, revealing 68 and 96 fecal bacteria-VOC associations at the family and genus level, respectively, that distinguish NA from HRA endpoints. For example, isobutyric acid associations with *Lachnospiraceae incertae sedis* and *Bacteroides* genera exhibit positive and negative regression lines for NA and HRA endpoints, respectively. However, the same chemical associates with *Coprococcus* and *Colinsella* genera exhibit the reverse regression line trends. Thus, fecal microbiota and VOC profiles and their associations in NA versus HRA individuals indicate the significance of multiple levels of analysis towards the identification of testable CRC risk biomarkers.

## 1. Introduction

Colorectal cancer (CRC) is the third most common and the second deadliest cancer worldwide, with 1.9 million new cases and 0.9 million recorded deaths in 2020 [1]. The multifactorial nature of CRC involves many risk factors, some of which have a clear environmental component and are thus modifiable, such as lifestyle, obesity, diet, alcohol intake, tobacco use, and biological aging, while others have a clear genetic component and are relatively fixed, such as, sex, ancestry, identifiable inherited mutations, and family history of proneness to cancer [2]. Two biomedically quantifiable and modifiable factors are the billions of microbes residing in the intestine and the thousands of metabolites they generate. These affect epithelial homeostasis and the host immune system and, in turn, tissue regeneration and predisposition to cancer [3,4]. Some gut microbes can ferment plant-derived dietary fibers and animal-protein-derived amino acids, facilitating host metabolism and a balanced intestinal biochemistry [5]. An emerging risk factor for CRC development is intestinal dysbacteriosis, which results from the presence of certain bacteria, diets, lifestyles, and clinical pathologies [2]. Accordingly, the intestinal bacteriome and metabolome provide the potential to identify novel non-invasive biomarkers for colonic inflammatory disorders and CRC. *Fusobacterium nucleatum*, colibactin-positive (pks+) *Escherichia coli*, and enterotoxigenic *Bacteroides fragilis* (ETBF) have been causally linked to CRC, while the link to CRC of *Clostridium symbiosum*, *Enterococcus fecalis*, *Streptococcus bovis*, *Peptostreptococcus anaerobius*, *Parvimonas micra*, and *Porphyromonas* species remains to be established [6,7,8,9,10,11]. Higher levels of some of these species and strains may distinguish high-risk adenoma (HRA) and early-stage CRC patients from healthy no-adenoma (NA) individuals. Moreover, microbial biomarker discovery may be improved when combined with the characterization of the intestinal metabolome. For example, combining ultra-high-performance liquid chromatography–mass spectrometry (UHPLC-MS) with metagenomics data allowed the link of cholesteryl esters and sphingolipids as well as of *Fusobacterium, Parvimonas*, and *Staphylococcus* with CRC and provided combinatorial microbiome–metabolome analysis towards early disease diagnosis [7]. Intriguingly, breath, urine, and fecal volatile organic compounds (VOCs) provide an alternative and promising clinical approach to intestinal inflammation and early CRC diagnosis, despite the inadequate strength of evidence and differing analytical platforms [12,13].

Here, we link fecal bacteria at different taxonomic levels and fecal volatile compounds to HRA status by sampling and analyzing a Cypriot population. We used 16S rRNA sequencing (16S-Seq) and headspace solid-phase micro-extraction gas chromatography–mass spectrometry (HS-SPME-GC-MS) to identify differences in fecal bacteria abundance and VOC concentrations between NA and HRA individuals. Moreover, we performed binomial logistic regression modeling to reveal fecal bacteria-VOC associations distinguishing NA from HRA endpoints.

## 2. Materials and Methods

### 2.1. Sample Collection

Fecal samples in this study were collected under the Cyprus Intestinal Health Study (MoCo Project EXCELLENCE/1216/0523) funded by the Research and Innovation Foundation of Cyprus. Bioethical approval was obtained from the Cyprus National Bioethics Committee (Protocol numbers: EEBK/ΕΠ/2015/38 and EEBK/ΕΠ/2019/23). A total of 117 participants provided fecal samples ≥15 days after conventional colonoscopy per established clinical indications [14], which were stored at −80 °C until analysis. In total, 100 of these were assigned an NA status due to the absence of tumor detection during colonoscopy. The remaining 17 individuals were assigned an HRA status according to colonoscopy and histopathological reports recording ≥3 adenomas/serrated polyps, or ≥1 adenoma/serrated polyp ≥1 cm, or ≥1 villus or tubulovillus adenoma, or high-grade dysplasia per established criteria [15,16]. In total, 55 and 62 individuals were males and females, respectively, 50–70 years old, undergoing routine colonoscopy (Table 1). No individual had a history of CRC or inflammatory bowel disease (IBD) or received antibiotics treatment or suffered from gastroenteritis during the month before colonoscopy or sample collection.

### 2.2. Fecal Bacteria DNA Isolation and 16S Gene Amplicon Sequencing

Fecal bacteria DNA isolation and purification was performed using the PureLink™ Microbiome DNA Purification Kit (Invitrogen™) using 0.18–0.2 g from the initial fecal sample, and 90 μL of eluted DNA was stored at −80 °C. The 16S rRNA gene hypervariable regions V3 and V4 were sequenced using the Nextera XT Library Preparation kit (Illumina™, Inc., San Diego, CA, USA) [17] and the following primers containing overhang adapter sequences:

Forward: TCGTCGGCAGCGTCAGATGTGTATAAGAGACAGCCTACGGGNGGCWGCAG

Reverse: GTCTCGTGGGCTCGGAGATGTGTATAAGAGACAGGACTACHVGGGTATCTAATCC

16S Gene Amplicon Sequencing for the taxonomic classification was performed using a 300 bp paired-end run on an Illumina MiSeq™ platform, following the standard Illumina protocols.

### 2.3. Bioinformatics and Statistical Analyses for Metagenomics

Metagenomics Operational Taxonomic Units (OTU) analysis was performed, determining the relative abundance of each bacterial taxon from phylum to species. Bioinformatic OTU analysis was performed from FASTQ files with paired-end reads utilizing Ribosomal Database Project Classifier against the RefSeq RDP 16S ver4.3 database [18]. Quality control was applied by requiring the detection of each OTU in at least 30% of samples and a minimum average relative abundance of 1% in at least one of the two groups (NA and HRA). Average bacterial abundance at the phylum, family, genus, and species taxonomic level was compared between the NA and HRA groups using a Mann–Whitney U test.

To display bacterial taxa percentiles (raw relative abundance) and hierarchy based on identified reads for the NA and the HRA groups, we used Krona visualization with classification imported in an Excel template detailing lineage and magnitude [19]. Inverse Simpson index was used to calculate alpha diversity and UniFrac index to compute beta diversity, both using the VEGAN package [20].

### 2.4. HS-SPME-GC-MS Headspace Analysis of Fecal VOCs

A total of 18 NA and 10 HRA individuals were randomly selected for fecal volatolomics. Approximately 0.6 g of each frozen sample added in a 20 mL headspace glass vial (Agilent; Part#: 8010-0413) was thawed for 24 hrs at room temperature. Then, 5 μL of internal standard solution of chlorobenzene-d5 (Sigma-Aldrich, Taufkirchen, Germany; Product#: 48086, CAS#: 3114-55-4) with a final concentration of 25 ppb was injected in the headspace vial the sample left to equilibrate in the closed crimp seal vial for 24 h at room temperature and it was then incubated in a water bath at 60 °C for 1 h. Consecutively, the 75 μm CAR/PDMS SPME fiber was exposed to the headspace phase of the vial for 30 min, so as to achieve the extraction of the small volatiles contained in the headspace phase. VOCs were thermally desorbed from the SPME fiber in an Agilent Single-Quadrupole GC-MS Instrument (GC-7890B, MSD-5977B, Agilent Technologies, Santa Clara, CA, USA) (Appendix A).

### 2.5. Volatolomic and Combinatorial Omic Statistical Analyses

For normalization, raw VOC values were divided by the value of the internal standard (chlorobenzene-d5, 25 ppb). Quality control was applied by minimal detection of at least 3 values of VOCs in both groups. To assess VOC abundance differences between NA and HRA groups, the Shapiro–Wilk normality test was performed prior to the Mann–Whitney U test (non-parametric, if normality fails), or independent sample *t*-test (parametric, for normally distributed variables). Principal component analysis (PCA) was conducted with the FACTOEXTRA package in R and the production of a heatmap to show the values of VOCs across each sample was performed with the GPLOTS package in R [21,22]. The probability of associations of VOCs with bacterial families or genera relative abundance in predicting NA from HRA status was tested via binomial logistic regression analysis, using the R libraries ggplot2, sjPlot, scales, grid, gridExtra, writexl, ggpubr, factoextra, dplyr, tidyr, readxl, ggstatsplot, vegan, agricolae, and circlize. The a priori threshold for statistical significance in all tests was set at *p*-value ≤ 0.05.

## 3. Results

### 3.1. Fecal Bacteria Prevalence and Diversity in NA and HRA Individuals

Fecal samples from 117 female and male adults between the ages of 50 and 70 (Table 1), divided into 100 NA and 17 HRA in accordance with established criteria [15], were analyzed via 16S-Seq, generating 38.6 million quality-filtered reads, 87% of which were identified. Krona plots revealed raw relative abundances as percentiles of total identified bacteria sequence reads per taxonomic level for the NA and the HRA group (Figure 1 and Figure 2). Krona plots display the percentiles of classified sequence reads found in all samples of each group and thus comparisons between the NA and HRA groups are—unlike those in all of the following results sections—not subject to statistical evaluation.

Indicatively, Firmicutes was the most prevalent phylum in both groups, covering 68% in NA and 41% in HRA individuals. *Lachnospiraceae* and *Ruminococcaceae*, the most prominent Firmicutes families in the NA group, were tentatively less prevalent in the HRA group. The *Blautia*, *Roseburia*, and *Fusicatenibacter* genera collectively covered 48% of the *Lachnospiraceae* family sequence reads in ΝA, and 19% in HRA individuals. Bacteroidetes and Actinobacteria phyla were comparable in the two groups: Bacteroidetes covered 20% of the sequence reads in NA, and 14% in HRA individuals; while Actinobacteria covered 9% in NA and 10% in HRA. Interestingly, Proteobacteria covered 23% and 1% of the sequence reads in HRA and NA individuals, respectively. Accordingly, *Enterobacteriaceae,* a prominent Proteobacteria family, was prominently abundant in HRA individuals.

The inverse Simpson index was applied to assess alpha diversity within the groups of 100 NA and 17 HRA individuals at the phylum, family, genus, and species level. The higher the value of this index, the greater the diversity within the group. As expected, the alpha diversity increased for each group from family to species level, but the index of the NA and HRA groups at a given taxonomic level was in all cases comparable (Appendix A). To determine potential dissimilarities in the microbial communities between the NA and the HRA group, the phylogenetic distance between sets of phyla, families, genera, and species unique to either the NA or the HRA group we used the UniFrac phylogenetic method. The Multidimensional Scaling (MDS) representation of beta-diversity measurement showed no significant differences at any taxonomic level between the NA and HRA groups (Appendix A).

### 3.2. Significant Fecal Bacteria Differences between NA and HRA Individuals

We used a Mann–Whitney non-parametric test to pinpoint statistically significant differences between NA and HRA individuals in the relative abundance of bacteria at the phylum, family, genus, and species level (Table 2, Figure 3). Differences were accepted at *p*-value ≤ 0.05 and normalized mean relative abundance ≥1% in at least one of the two groups. Accordingly, the phylum of Proteobacteria was ≈2 times more abundant in HRA than in NA individuals. Members of the *Lachnospiraceae* family, namely, the *Roseburia* and *Fusicatenibacter* genera, were ≈2 times more abundant in NA individuals. At the species level, *Roseburia faecis, Blautia luti, Fusicatenibacter saccharivorans*, and *Eubacterium rectale* belonging to the *Lachnospiraceae* family, as well as *Phascolarctobacterium faecium* belonging to the *Acidaminococcaceae* family, were more abundant in NA individuals. In contrast, *Parabacteroides distasonis*, a bacterial species with a potential pathogenic role belonging to the *Tannerellaceae* family, was ≈3 times more abundant in HRA individuals.

### 3.3. VOC Abundance in NA and HRA Individuals

HS-SPME-GC-MS analysis was used to evaluate the profile of volatile organic compounds in the feces of a subset of the initial individuals: 18 NA and 10 HRA individuals. Out of over 250 detected volatiles, 71 were present in the samples of at least three individuals in each group. Principal component analysis (PCA) for the 71 essential volatiles emitted (Appendix A) indicated dispersed distributions of samples, that is, significant sample-to-sample variations in VOCs. Moreover, there was a lack of distinct sample clustering, indicating similarity between the VOCs of the NA and HRA groups. This may be partially due to differences in the dietary habits of the sampled individuals. However, branched-chain fatty acids (BCFAs), methyl propionate, methyl butyrate, and isobutyric acid, and the ketones, methyl butyl ketone (2-Hexanone) and ethyl acetone (2-Pentanone), were significantly more abundant in HRA than in NA individuals (Figure 4, Appendix A).

### 3.4. Associations between Bacterial Families and VOCs in NA and HRA Individuals

We examined associations between fecal bacterial families (FBFs) and fecal VOCs using 28 fecal samples able to distinguish 18 NA from 10 HRA endpoints using binomial logistic regression. We identified 68 pairwise associations between 16 FBFs and 30 VOCs, as shown in the Chord diagram of Figure 5 and Appendix A. Of all FBFs, *Bacteroidaceae* and *Eubacteriaceae* exhibited the most associations with VOCs, 10 and 9, respectively. Of all VOCs, acetaldehyde and propanal exhibited the most associations with FBFs, seven of them each.

The 68 associations include four VOCs enriched in HRA individuals, butyl methyl ketone, isobutyric acid, methyl butyrate, and ethyl acetone (Figure 4), which are associated with six FBFs, *Eubacteriaceae, Lactobacillaceae, Bacteroidaceae, Erysipelotrichaceae, Acidaminococcaceae,* and *Peptostreptococcaceae* (Table 3 and Figure 5), and *Lachnospiraceae*, an FBF enriched in NA individuals (Figure 3), which is associated with four VOCs, propanal, methacrolein, methyl 4-methylvalerate, and dimethyl trisulfide (Table 3, Figure 5). Indicative of discrimination between NA and HRA, the regression line slopes for the NA and the HRA endpoints was opposite for most of these FBF-VOC associations (Table 3, Figure 6).

### 3.5. Associations between Bacterial Genera and VOCs in NA and HRA Individuals

We also examined associations between fecal bacterial genera (FBGs) with fecal VOCs using the 28 fecal samples from 18 NA and 10 HRA individuals using binomial logistic regression. We identified 96 pairwise associations between 27 fecal bacterial genera and 41 VOCs, as shown in the Chord diagram of Figure 7 and Appendix A. Of all FBGs, *Bacteroides* and *Eubacterium* exhibited the most associations with VOCs, 10 and 9, respectively. Of all VOCs, propanal exhibited the most (13) associations with FBGs, while acetaldehyde, isobutyraldehyde, isovaleraldehyde, and methyl cyclohexanecarboxylate associated with five FBGs each.

The 96 associations include four VOCs enriched in HRA individuals, butyl methyl ketone, isobutyric acid, methyl butyrate, and ethyl acetone (Figure 4), which are associated with eight FBGs, *Ruminococcus, Lachnospiraceae incertae sedis, Collinsella, Bacteroides, Coprococcus, Bacteroides, Holdemanella*, and *Eubacterium*, and two FBGs enriched in NA individuals, *Roseburia* and *Fusicatenibacter* (Figure 3), which are associated with three VOCs, propanal, p-Cresol, and indole (Table 4, Figure 7). Indicative of discrimination between NA and HRA, the regression line slopes for the NA and the HRA endpoints was opposite for most of these FBG–VOC associations (Table 4, Figure 8).

## 4. Discussion

Countless interactions take place within the large intestine between volatile compounds and the intestinal microbiota influenced by host diet, age, metabolism, inflammation-related processes, medications, and other environmental factors. VOCs are generated, modified, or degraded by bacteria residing the intestine and the host itself. Some VOCs may benefit, while others may destroy, the health-promoting microbial composition balance [31]. During the last decade, microbiome changes and specific bacterial species have been linked to cancer development and progression [32,33]. Nevertheless, it is still early to name microbial and biochemical signatures of predictive value for CRC risk. Through our ongoing clinical study, we find significant changes in the microbial composition and fecal VOCs between NA and HRA individuals residing in the island of Cyprus. Beyond our independent microbiome and volatilome analysis, we indicate a level of complexity revealed by assessing fecal bacteria–VOC associations, and another one by assessing differences in these associations between NA and HRA individuals.

### 4.1. Fecal Bacteriome Analysis

To determine differential microbial abundances between the NA and HRA individuals, we compared microbial composition at the phylum, family, genus, and species level utilizing Mann–Whitney non-parametric statistical analysis. Proteobacteria and *Parabacteroides distasonis*, a member of the *Tannerellaceae* family, were more abundant in HRA individuals, while members of the *Lachnospiraceae* family, *Roseburia faecis, Blautia luti, Fusicatenibacter saccharivorans*, and *Eubacterium rectale*, as well as *Phascolarctobacterium faecium*, a member of the *Acidaminococcaceae* family, were more abundant in NA individuals.

Based on previous studies, depletion of members of the Clostridia class, such as *Lachnospiraceae, Ruminococcaceae*, and *Eubacteriaceae*, indicates colonic subclinical inflammation within which high-risk adenomas may form [2,34]. Clostridia can ferment dietary plant fibers producing butyrate and other SCFAs, such as propionate, acetate, and valerate. Moreover, the anaerobic *Lachnospiraceae* and *Ruminococcaceae* families may play a preventive role in CRC development, since the relative abundances of *Lachnospiraceae* and its metabolites have been inversely correlated with CRC progression [24].

*R. faecis, B. luti, E. rectale*, and *F. saccharivorans* species are members of the *Lachnospiraceae* family and are SCFA producers that preserve gut homeostasis and protect the intestinal mucosal cells from becoming hyperplastic, dysplastic, or malignant by regulating colonic inflammation [35]. Moreover, members of the *Fusicatenibacter* and *Roseburia* genera, such as *F. saccharivorans* and *R. faecis*, are significantly reduced in CRC patients compared to healthy individuals [23,30]. Similarly, *B. luti* is depleted in CRC patients vs. healthy controls [25,36], consistent with its probiotic properties [27]. *P. faecium*, another SCFA producer, utilizes succinate, generated in the large intestine by bacteria of the *Bacteroides* and *Parabacteroides* genera, sustaining its abundance during aging [37]. The abundance of the five aforementioned commensal bacterial species may thus indicate healthy versus cancer-prone status and may serve as bacterial biomarkers of health, although prospective and experimental studies are required to provide deeper insight into their role.

On the other hand, *P. distasonis* appears to be a pathobiont in some cases, present in a healthy gut, while enriched in human abscesses, extra-intestinal abdominal infections, and in Lynch syndrome patients [38,39,40,41]. While potentially anti-inflammatory and protective against CRC in other cases, it is associated with pre-existing inflammatory bowel disease in both humans and animal models [38]. Therefore, *P. distasonis* along with Proteobacteria species being enriched in HRA individuals may be indicative of intestinal inflammation and CRC risk.

Raw relative abundance of *Gammaproteobacteria* class and the *Enterobacteriaceae* family derived from Krona plots was many-fold higher in HRA individuals, in agreement with their clear association with conventional and serrated adenomas [34,42]. Similarly, the low but detectable presence of the opportunistic pathogen and oncobacterium *Fusobacterium nucleatum* was only recorded in HRA individuals, indicating potential similarities between the microbial ecosystem composition within the gut of HRA individuals and CRC patients [43,44,45,46,47].

### 4.2. Fecal Volatilome Analysis

Methyl propionate, a carboxylic ester, and methyl butyrate, a fatty acid ester, were found elevated in the feces of HRA individuals, suggesting that these BCFAs may contribute to pathogenesis [48]. Both of them are low-molecular-weight volatiles, highly abundant in the human feces, and products of the exogenously esterification of propionate and butyrate derived from dietary fibers and microbial metabolism within the large intestine [5,49].

Isobutyric acid is a branched-chain saturated fatty acid primarily derived from the branched-chain amino acid valine via intestinal fermentation mediated by the *Clostridium* and *Bacteroides* species. Its concentration increases progressively along the proximodistal colon axis and in feces. BCFAs are proposed to affect human health but they are relatively unexplored compared to SCFAs [5]. Interestingly, we found isobutyric acid and *Bacteroides* and *Clostridium* genera in higher levels in the feces of HRA individuals. Similarly, we found methyl propionate, methyl butyrate, and isobutyric acid in higher levels in HRA individuals, in agreement with previous findings about the greater abundance of isobutyric acid in CRC relative to HRA and healthy control individuals [50]. These BCFAs may thus serve as candidate biomarkers of CRC risk.

### 4.3. Fecal Bacteriome to Volatilome Analysis

Binary logistic regression analysis of fecal bacteria families and genera with fecal VOCs revealed 68 FBF–VOC associations (Figure 5), 11 of which pertain to HRA-enriched VOCs and NA-enriched FBFs (Table 3, Figure 6), and 96 FBG–VOC associations (Figure 7), 12 of which pertain to HRA-enriched VOCs and NA-enriched FBGs (Table 4, Figure 8). Isobutyric acid associates with the *Lacnospiraceae incerta sedis*, *Bacteroides, Colinsella*, and *Coprococcus* genera in different ways between the NA and HRA groups (Table 4). The association of this BCFA with *Lacnospiraceae incerta sedis* and *Bacteroides* exhibited a positive regression line slope in NA individuals, and a negative one in HRA individuals (Table 4, Figure 8). However, isobutyric acid associations with *Coprococcus* and *Colinsella* exhibited regression line slopes in reverse: negative in NA and positive in HRA individuals (Table 4, Figure 8). BCFAs, such as isobutyric and isovaleric acids, are less abundant than SCFAs in the human large intestine and feces. They are markers of amino acid fermentation, and their intestinal and fecal abundance is related to diet and aging [5]. Previous studies indicate that members of the Proteobacteria and Actinobacteria phyla, the *Lachnospiraceae* family, and *Fusobacteria* and *Bacteroides* genera, are main producers of BCFAs and markers of colonic protein fermentation, a process that also generates p-cresol, phenol, and ammonia [5,51,52]. Moreover, we associated *Lachnospiraceae* with four VOCs, propanal, methacrolein, methyl 4-methylvalerate, and dimethyl trisulfide (Table 3). Propanal was also associated with *Roseburia* (Table 4), but the biological significance of the association of this and other potentially pathogenic VOCs with potentially beneficial bacteria is unclear due to the positive correlation of this volatile in breath (not fecal) samples of CRC patients [53]. Notably, breath-sample VOCs are expected to defer from fecal VOCs, despite their potential predictive power [54,55].

Methyl butyrate, a volatile compound involved in SCFA butyrate production, was associated with *Bacteroides* and *Coprococcus* genera (Table 4). Moreover, it was positively associated with *Bacteroides* in NA individuals and negatively in HRA individuals (Table 4, Figure 8). On the other hand, methyl butyrate was negatively associated with *Coprococcus* in NA individuals and positively in HRA individuals (Table 4, Figure 8). These observations need to be further investigated, since *Bacteroides* and *Coprococcus* genera are abundant SCFA producers and regulators of protein fermentation and complex oligosaccharides digestion within the gut [51,56].

We also highlight ethyl acetone (2-hexanone) as highly abundant in the human feces [57], which was positively associated with *Eubacterium* in NA individuals, while negatively in HRA individuals (Table 4, Figure 8). The depletion of *Eubacterium* is noted in IBD individuals and those adopting a western diet of high intake of animal protein and fat and less intake of plant fiber [58]. Furthermore, ethyl acetone was positively associated with *Holdemanella*, an *Erysipelotrichaceae* family member, in NA individuals, while negatively with HRAs. Both associations need to be further investigated, since *Holdemanella biformis* has been suggested as an antitumorigenic SCFA generator able to control CRC cell proliferation and intestinal metabolism [59,60,61].

P-cresol and indole were detected in all fecal samples tested, since they are the main fermentation products of the essential amino acids, phenylalanine and tyrosine (p-cresol), and tryptophan (indole). Both volatiles are also products of bacterial metabolism within the large intestine, and precursors of toxic metabolite compounds, referred to as uremic toxins; p-cresol has the potential to contribute to genotoxicity and colorectal oncogenesis. Their concentration among fecal samples varies widely, as a result of dietary differences and protein intake from animal versus plant sources [62]. Both p-cresol and indole were positively associated with *Fusicatenibacter* abundance in both subgroups of NA and HRA subjects, although p-cresol was overall more abundant than indole. This agrees with previous studies showing members of the *Lachnospiraceae*, *Clostridiaceae*, *Eubacteriaceae*, *Peptostreptococcaceae*, *Enterobacteriaceae*, *Oscillospiraceae*, and *Sutterellaceae* families produce p-cresol and indole in culture [63,64], and associate with p-cresol and indole in human feces [62].

Despite the wealth of bacteriome and volatile correlations, no combinatorial biomarkers of CRC risk have been established [65,66,67]. Thus, microbial–metabolite signatures need to be further investigated to address the potentially toxic volatiles differentially produced by the dysbiotic microbiome of HRA individuals. One limitation of this and previous studies is that metagenomics and volatolomics analyses used fecal samples, reflecting the relative abundances of bacterial communities and VOCs at the lumen of the distal GI tract, without taking into consideration the other parts of the colon or the mucosa. Different levels of microbial abundances and volatile emissions derived in fecal matter cannot directly reflect the complex host–microbiota interactions taking place within the colonic mucosa of the proximal and distal colon. Hence, studies sampling different sites along the colonic mucosa may provide a broader picture of the metagenomic and metabolomic milieu.

## 5. Conclusions

Our study links fecal bacteria at different taxonomic levels, fecal volatile compounds, and their associations to the HRA status of participants. Unlike correlation analysis merely linking bacteria to metabolites irrespective of health status, our binomial logistic regression modeling focused on fecal bacteria–VOC associations, distinguishing NA from HRA endpoints. As such, it provides an additional level of participant stratification by revealing metabolite–bacteria associations that are biased according to NA versus HRA status. Following a similar path, future studies aiming to establish biomarkers of HRA will benefit from including more participants and colon site-specific sampling.

## Figures and Tables

**Figure 1 metabolites-13-00819-f001:**
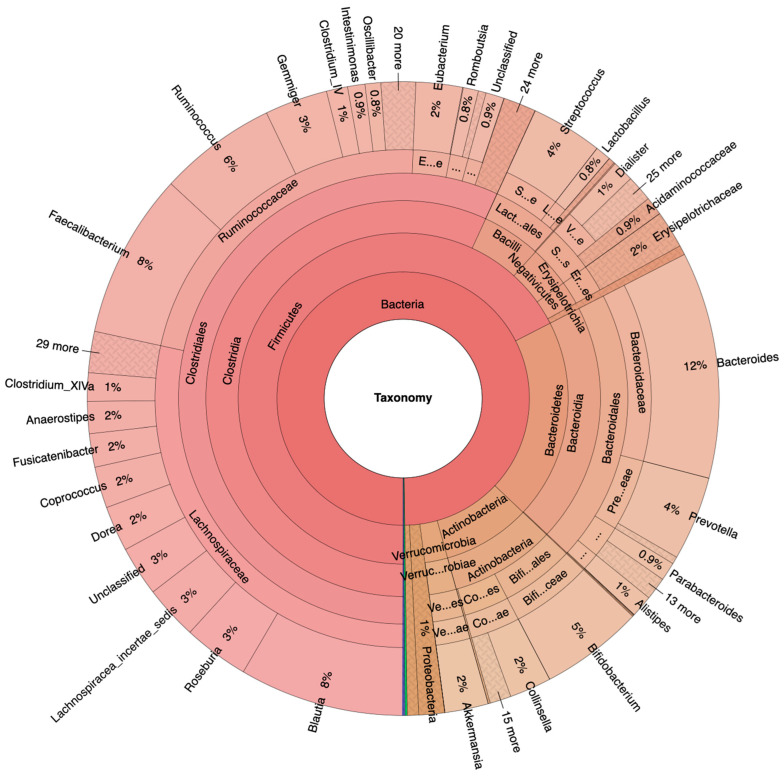
Krona plots of NA (https://www.stremble.com/papersuplements/KronaGroupHealthy.html, accessed 15 May 2023) individuals showing the percentile of identified sequence reads and their phylum to genus hierarchy.

**Figure 2 metabolites-13-00819-f002:**
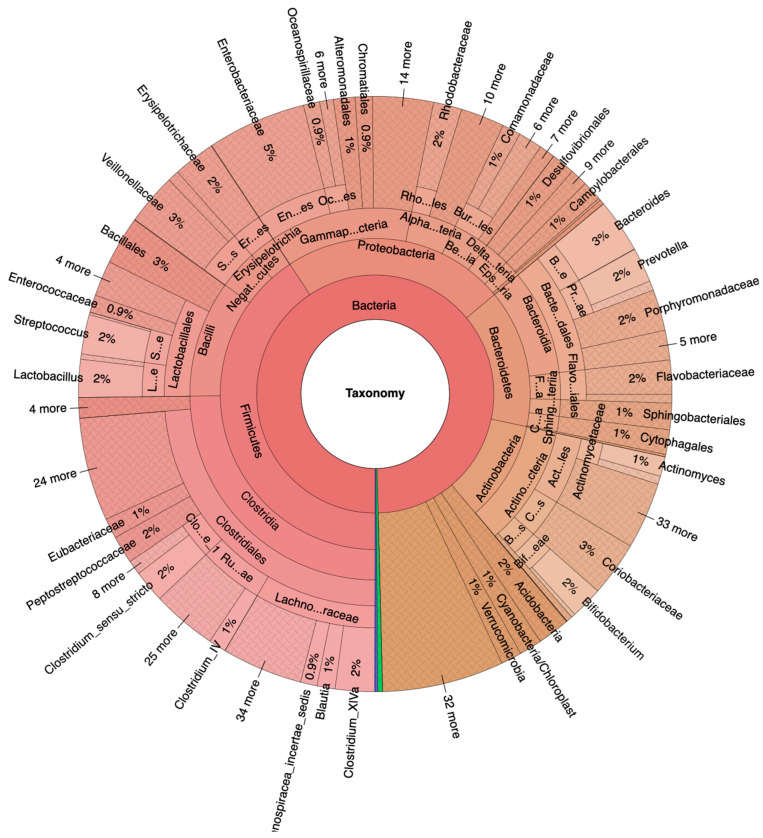
Krona plots of HRA (https://www.stremble.com/papersuplements/KronaGroupCancerProne.html, accessed 15 May 2023) individuals showing the percentile of identified sequence reads and their phylum to genus hierarchy.

**Figure 3 metabolites-13-00819-f003:**
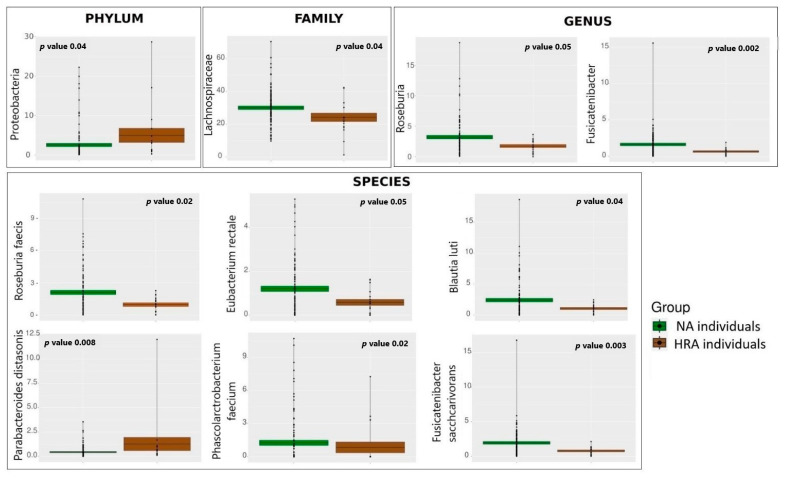
Average relative abundance of specific bacterial taxa displaying significant differences between the NA (green) and the HRA (brown) individuals. The mean is indicated by the horizontal line dividing each box in two. The top and the bottom of each box indicate the mean plus/minus the standard error. Dots display data points and *p*-value indicates the significance of the difference between NA and HRA.

**Figure 4 metabolites-13-00819-f004:**
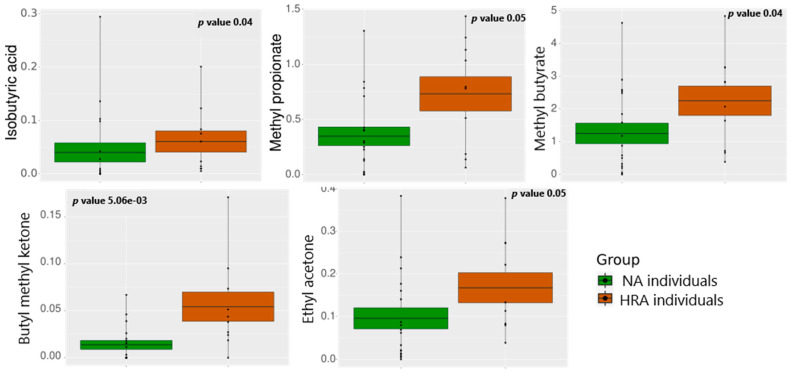
Relative abundance of VOCs displaying significant differences between the NA (green) and the HRA (brown) group. The mean is indicated by the horizontal line dividing each box in two. The top and the bottom of the box indicate the mean plus/minus the standard error. Dots display data points.

**Figure 5 metabolites-13-00819-f005:**
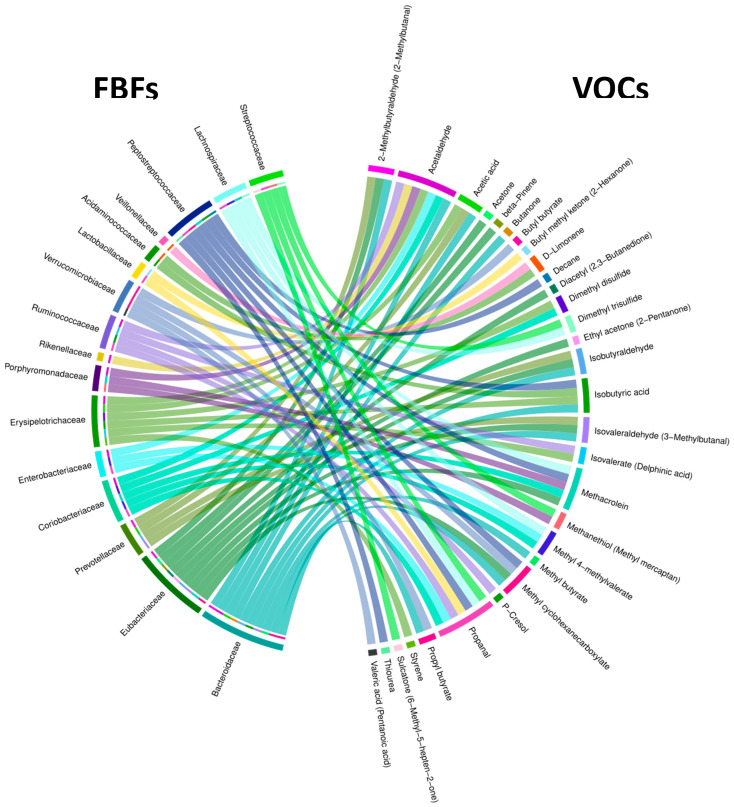
Chord diagram showing all 68 pairwise FBF-VOC associations discriminating NA from HRA endpoints. The outer ring shows FBFs on the left and VOCs on the right. The inner half ring on the left side and the ribbons spanning the circle side to side show specific FBFs interacting with specific VOCs.

**Figure 6 metabolites-13-00819-f006:**
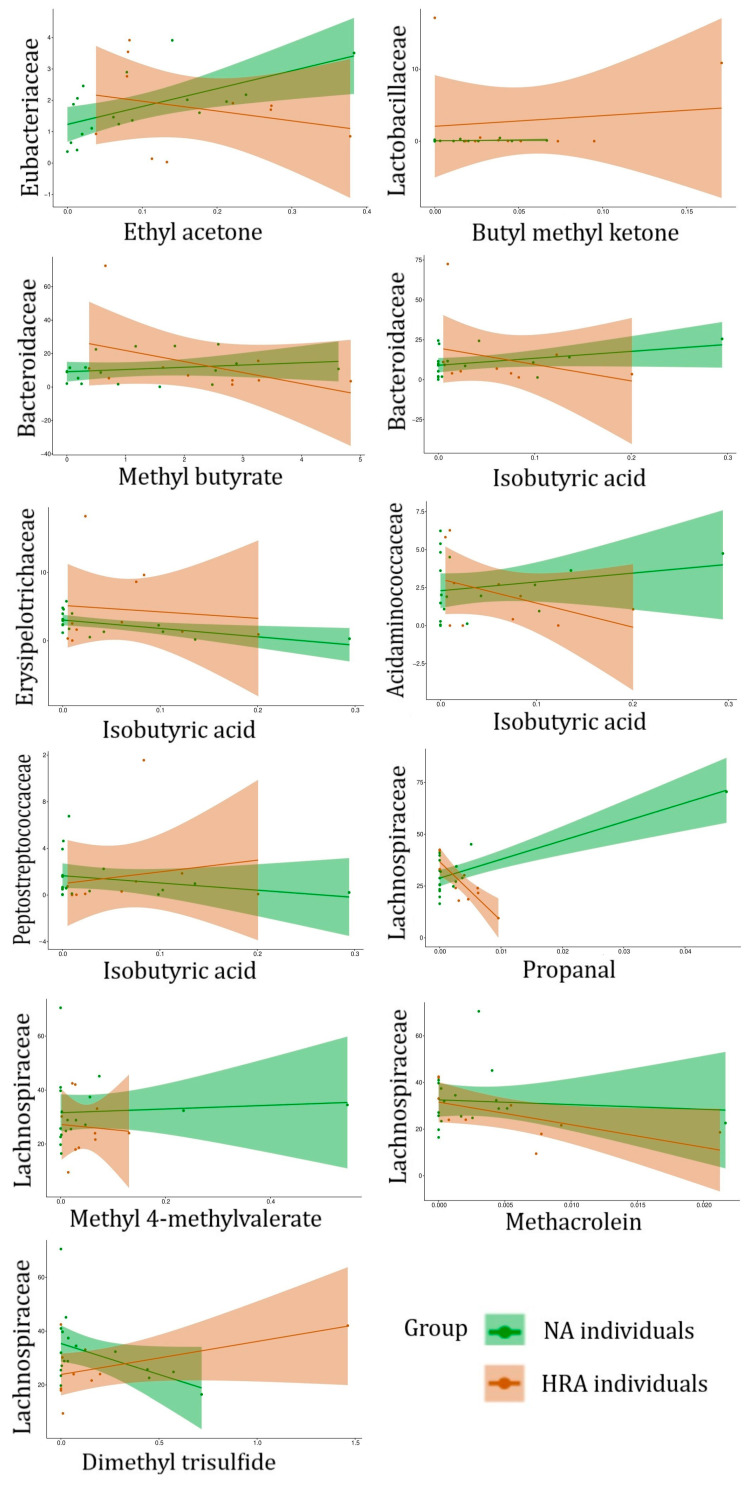
Regression lines and confidence intervals of NA versus HRA endpoints of FBF–VOC associations shown in Table 3. Relative abundance of FBFs (on the *y*-axis) against the concentration of the VOCs associated with them (on the *x*-axis) using 18 NA and 10 HRA endpoints, to forecast the probability of the individual’s outcome (NA/healthy or HRA/CRC prone). Regression lines and confidence intervals are displayed in green and red for NA and HRA endpoints, respectively. Only HRA-enriched FBFs or NA-enriched VOCs are displayed.

**Figure 7 metabolites-13-00819-f007:**
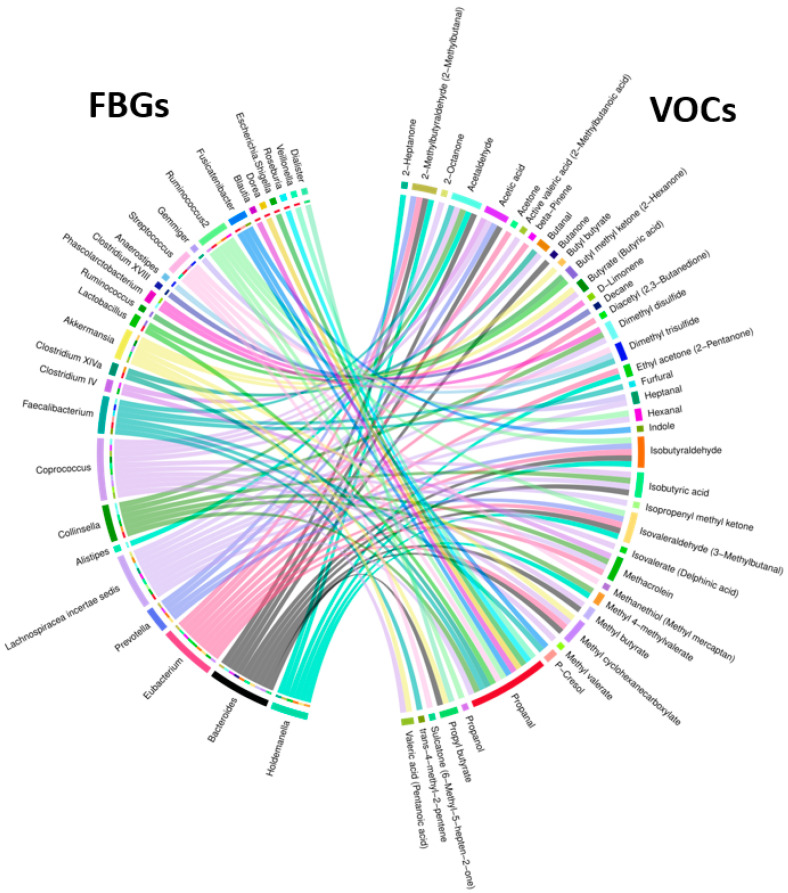
Chord diagram showing all 96 pairwise associations between FBGs and VOCs discriminating NA from HRA endpoints. The outer ring shows FBGs on the left and VOCs on the right. The inner half ring on the left side and the ribbons spanning the circle side to side show specific FBGs interacting with specific VOCs.

**Figure 8 metabolites-13-00819-f008:**
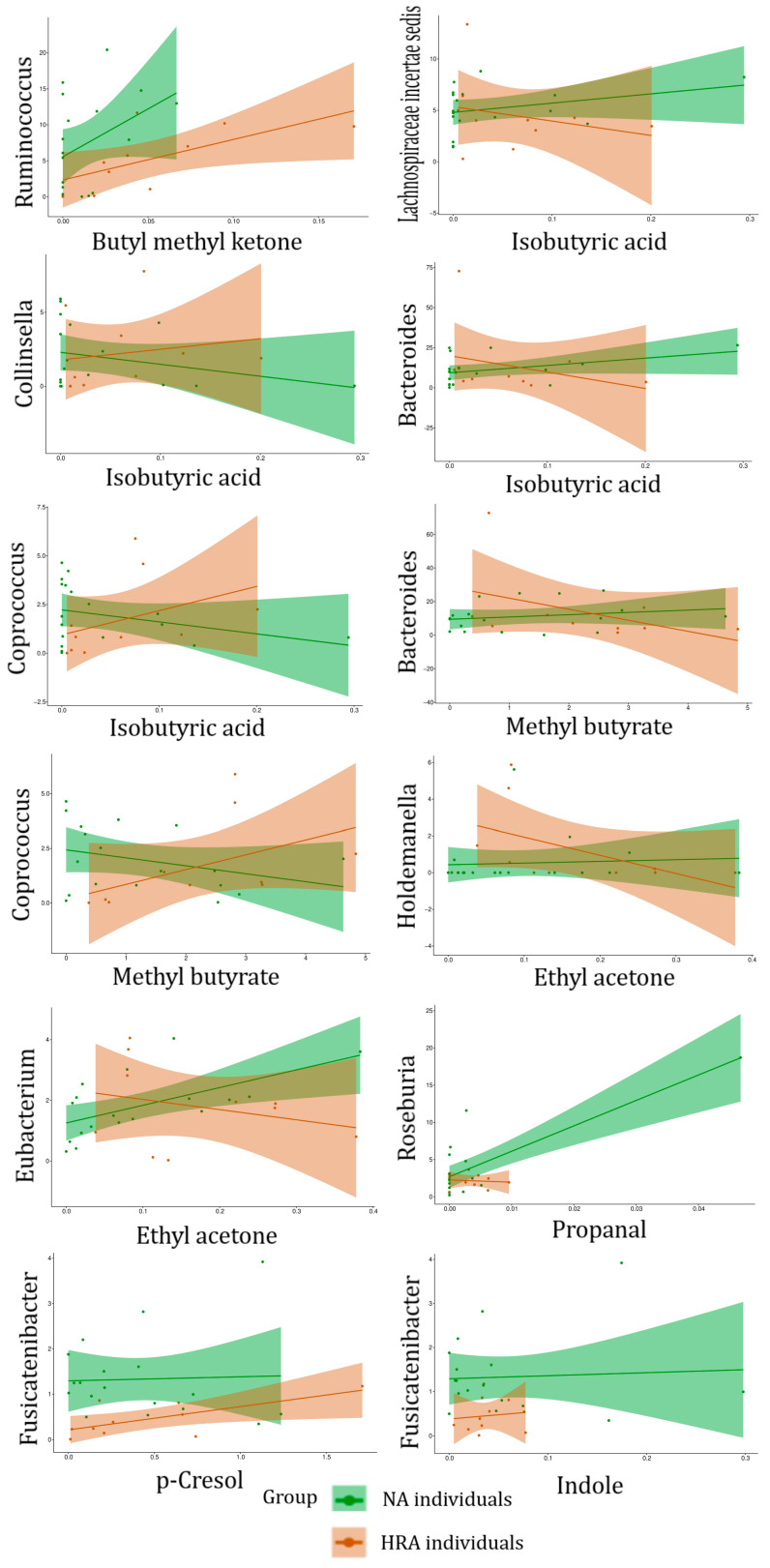
Regression lines and confidence intervals of NA versus HRA endpoints of FBG-VOC associations shown in Table 4. Relative abundance of FBGs (on the *y*-axis) against the concentration the VOCs associated with them (on the *x*-axis) using 18 NA and 10 HRA endpoints, to forecast the probability of the individual’s outcome (NA/healthy or HRA/CRC prone). Regression lines and confidence intervals are displayed in green and red for NA and HRA data sets, respectively. Only HRA-enriched FBGs and NA-enriched VOCs are displayed.

**Table 1 metabolites-13-00819-t001:** Participant breakdown per sex, age bracket (in years), assay (16S-Seq vs. GC-MS), and macroscopic classification (NA vs. HRA).

Participants per Assay and NA/HRA Status	Sex		Age Bracket		
	*Females*	*Males*	*50–60*	*61–65*	*66–70*
All 117 individuals	62 (53%)	55 (47%)	32 (27%)	43 (37%)	42 (36%)
16S-Seq: 100 NA individuals (85.5%)	52	48	27	36	37
16S-Seq: 17 HRA individuals (14.5%)	10	7	5	7	5
GC-MS: 18 NA individuals (15.4%)	10	8	5	7	6
GC-MS: 10 HRA individuals (8.5%)	4	6	3	4	3

**Table 2 metabolites-13-00819-t002:** Significant differences in the relative abundance between 100 NA and 17 HRA individuals at the phylum, family, genus, and species level. Average, median, and 1st and 3rd quartile percentiles are displayed.

	Organism *p-Value*	Average, Median, and IQR (Q1–Q3) of Relative Abundance (%) in HRA	Average, Median, and IQR (Q1–Q3) of Relative Abundance (%) in NA	Potential IMPACT on the Host
**PHYLUM**	Proteobacteria *p = 0.0440*	**4.96** 2.17, 1.20–5.26	2.53 0.94, 0.60–2.15	Potential pathogens, such as *E. coli, Salmonella, Vibrio cholerae*, and *Helicobacter pylori*. Infectious, inhibit immune function, cause dysbacteriosis, and exacerbate growth of colon cancer cells [23].
**FAMILY**	*Lachnospiraceae* *p = 0.0393*	24.26 23.99, 18.63–30.29	**30.09** 29.02, 24.53–34.90	Beneficial. Protect against colon cancer by producing butyrate via the butyrate kinase pathway [24].
**GENUS**	*Roseburia* *p = 0.0481*	1.76 2.42, 1.22–2.87	**3.23** 2.40, 1.33–3.99	Beneficial. SCFA producers protecting against gut inflammation, maintaining energy homeostasis, inhibiting NF-κB activation.
	*Fusicatenibacter* *p = 0.0025*	0.67 0.79, 0.24–0.89	**1.65** 1.30, 0.60–2.28	Beneficial. Butyrate producers maintaining intestinal regeneration, homeostasis, low inflammation [25].
**SPECIES**	*Roseburia faecis* *p = 0.0189*	0.96 0.93, 0.31–1.39	**2.10** 1.52, 0.68–2.75	Beneficial. SCFA producer [26].
	*Blautia luti* *p = 0.0423*	1.06 1.03, 0.49–1.50	**2.41** 1.42, 0.75–2.81	Beneficial. Potential anti-inflammatory action and inhibition of pathogen colonization via production of bacteriocins [27].
	*Fusicatenibacter saccharivorans* *p = 0.0030*	0.79 0.92, 0.27–1.10	**1.94** 1.49, 0.72–2.74	Beneficial. Butyrate producer decreased in the gut of CRC patients [23].
	*Eubacterium rectale* *p = 0.0456*	0.58 0.52, 0.07–0.84	**1.12** 0.78, 0.24–1.74	Beneficial. Butyrate producer [28].
	*Phascolarctobacterium faecium* *p = 0.0164*	0.84 0.0025, 0.0–0.004	**1.25** 0.019, 0.002–1.45	Beneficial. Propionate producer via the succinate metabolic pathway [29].
	*Parabacteroides distasonis* *p = 0.0084*	**1.26** 0.57, 0.23–0.96	0.43 0.24, 0.05–0.49	Potentially pathogenic and carcinogenic, associated with CRC [30].

Higher average abundance is indicated in bold and the significance of difference with the *p*-value. IQR, interquartile range (Q1–Q3); SCFAs, short chain fatty acids. Percentiles of reads classified at each taxonomic level (mean, SEM): phylum (mean 99.68%, 0.00014), family (98.21%, 0.11), genus (94.91%, 0.21), species (80.76%, 0.578).

**Table 3 metabolites-13-00819-t003:** FBF–VOC associations of HRA-enriched VOCs and NA-enriched FBFs discriminating NA from HRA endpoints. Four VOCs enriched in HRA fecal samples (butyl methyl ketone, isobutyric acid, methyl butyrate and ethyl acetone) associated with six fecal bacterial families, and one fecal bacterial family enriched in NA fecal samples (*Lachnospiraceae*) associated with four VOCs. The significance of associations and the regression line slope (positive or negative) of NA versus HRA endpoints is indicated.

	FBFs	Associated VOC	Significance of Association, and Regression Line Slope for NA and HRA Endpoints
**FBFs associated with** **HRA-enriched VOCs**	*Eubacteriaceae*	Ethyl acetone (2-Pentanone)	*p = 0.02* NA (+) HRA (−)
	*Lactobacillaceae*	Butyl methyl ketone (2-Hexanone)	*p = 0.04* NA (+) HRA (+)
	*Bacteroidaceae*	Methyl butyrate	*p = 0.03* NA (+) HRA (−)
		Isobutyric acid	*p = 0.04* NA (+) HRA (−)
	*Erysipelotrichaceae*		*p = 0.03* NA (−) HRA (−)
	*Acidaminococcaceae*		*p = 0.03* NA (+) HRA (−)
	*Peptostreptococcaceae*		*p = 0.04* NA (−) HRA (+)
**VOCs associated with** **NA-enriched FBFs**	*Lachnospiraceae*	Propanal	*p = 0.007* NA (+) HRA (−)
		Methacrolein	*p = 0.04* NA (−) HRA (−)
		Methyl 4-methylvalerate	*p = 0.03* NA (+) HRA (−)
		Dimethyl trisulfide	*p = 0.04* NA (−) HRA (+)

**Table 4 metabolites-13-00819-t004:** FBG–VOC associations of HRA-enriched VOCs and NA-enriched FBGs discriminating NA from HRA endpoints. Four VOCs enriched in HRA fecal samples (butyl methyl ketone, isobutyric acid, methyl butyrate, and ethyl acetone) associate with seven fecal bacterial genera, and two FBGs enriched in NA fecal samples (*Roseburia* and *Fusicanibacter*) associate with three VOCs. The significance of associations and the regression line slope (positive or negative) of NA versus HRA endpoints is indicated.

	FBGs	Associated VOC	Significance of Association, and Regression Line Slope for NA and HRA Endpoints
**FBGs associated with** **HRA-enriched VOCs**	*Ruminococcus*	Butyl methyl ketone (2-Hexanone)	*p = 0.04* NA (+) HRA (+)
	*Lachnospiraceae incertae sedis*	Isobutyric acid	*p = 0.04* NA (+) HRA (−)
	*Collinsella*		*p = 0.05* NA (−) HRA (+)
	*Bacteroides*		*p = 0.04* NA (+) HRA (−)
	*Coprococcus*		*p = 0.004* NA (−) HRA (+)
	*Bacteroides*	Methyl butyrate	*p = 0.03* NA (+) HRA (−)
	*Coprococcus*		*p = 0.05* NA (−) HRA (+)
	*Holdemanella*	Ethyl acetone (2-Pentanone)	*p = 0.04* NA (+) HRA (−)
	*Eubacterium*		*p = 0.03* NA (+) HRA (−)
**VOCs associated with** **NA-enriched FBGs**	*Roseburia*	Propanal	*p = 0.02* NA (+) HRA (−)
	*Fusicatenibacter*	p-Cresol	*p = 0.03* NA (+) HRA (+)
		Indole	*p = 0.02* NA (+) HRA (+)

## Data Availability

The data presented in this study are available upon request from the corresponding author. The data are not publicly available due to bioethical protocol restrictions regarding participant privacy.

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
