# Peer review of "Fecal Microbiota and Associated Volatile Organic Compounds Distinguishing No-Adenoma from High-Risk Colon Adenoma Adults"

_metabolites, 2023, doi:10.3390/metabo13070819_

Round 1

Reviewer 1 Report

The manuscript of Katsaounou et al., show in a biostatistical  sound way the correlation of colonic microbiota analyzed by 16S amplicon sequencing and colonic volatiles by GS-MS with high-risk adenoma and non-adenoma. Moreover, they put their observations into perspective by linking establishing links with previously published information.  

Author Response

We thank the reviewer for appreciating the work and interpretation of our manuscript as is.

Reviewer 2 Report

The manuscript is very well written and provides an interesting perspective on the possible utilization of fecal microbiota and associated volatile organic compounds for monitoring risk of colon adenoma. 

Introduction of the manuscript is very objective and provides an interesting view of the state of the art in this field of research and highlights the needs for new approaches towards an effective early disease diagnosis.

Materials and methods are also very adequately described.

In these fields of research, we frequently encounter difficulties in establishing meaningful correlations in the results. This is not the case in this manuscript. The authors provide a very detailed and coherent presentation of results and outlined a rational strategy for deriving important correlations between fecal bacteria and detected volatile organic compounds.

Discussion of results is very objective and honest as well. The authors reach some relevant conclusions from their data analysis but point out from the start that “it is still very early to name microbial and biochemical signatures of predictive value for CRC risk”.

Minor comment:

i)                In the abstract the authors refer the number of participants (117) that went through routine colonoscopy and respective characterization of fecal microbiota but do not mention the number of participants that were used in the VOC analysis. In my opinion this information should be included in the abstract.

Author Response

We thank the reviewer for acknowledging the work described in our manuscript and its interpretation.

We agree with the comment provided by the reviewer. Accordingly we added (and marked in yellow) in the abstract the phrase "of samples from 28 participants" to mention the number of participants that were used in the VOC analysis.

Reviewer 3 Report

 The article concerns possible pathogenetic relations between changes in human fecal microbiota and spectrum of volatile substances produced in normal state and in subjects at high risk for colorectal cancer (HRA cohort). The design of study is quite rational.The samples obtained at sufficient terms (>15 days) after colonoscopy (a total of 117 cases) were subject to NGS for assessment of bacterial diversity, and to mass spectrometry of some volatile compounds produced by the microbiota. E.g., the authors  revealed higher concentrations of  isobutyric acid, methyl butyrate, methyl propionate, 2-hexanone and 25 2-pentanone in HRA persons, along with  lower abundance of Lachnospiraceae species, Roseburia faecis, Blautia luti, Fusicatenibacter saccharovorans, Eubacterium rectale and Phascolarctobacterium faecium.  The comparisons between controls and HRA are rational and convincing.  Meanwhile, some controversial changes were found when correlating the levels of metabolites produced in HRA and controls with abundance of distinct bacterial phyla/genera/species.

 Remarks:

 In Abstract:  -        The aim of this study should be clearly formulated (differences between HRA and controls, or correlations between microbiota diversity and production of volatile compounds) in the Abstract and Introduction.

-        Line 20-21:…. samples from 117 participants undergoing routine colonoscopy…   One should specify the time of fecal sampling …≥15 days after conventional colonoscopy (see Materials, line 82)… 

In Results:  -   Firmicutes seem to substantially prevail in normal subjects over HRA cases, when comparing Fig.1. and Fig.2. Is this difference significant?

-        The authors have shown that Proteobacteria, another dominant phylum, are less abundant in normal persons (as shown in Table 2). To justify  and characterize the intergroup differences, expecially, broad individual biodiversity of microbiota, one should show in Table 2 minimal and maximal values as well as median percentages for the bacteria of interest.

-        In Table 2, one should also mention the percentage of unclassified NGS reads when analyzing phyla, genera and species. These information losses are usually quite significant.

-        The numbers of HRA patients tested for volatile compounds (line 124: 18 normal versus 10 HRA subjects) seem to be rather small for reliable correlation analysis between highly variable microbiota diversity (especially, at species levels) and the spectrum of volatile compounds. Therefore, in  Table 3, one should add numerical values of the correlation quotients to make the strength of correlations more transparent.

In Conclusion, one should emphasize a preliminary character of this study, due to limited number of cases in correlation analysis. Meanwhile, the general idea of correlating intestinal microbiome and metabolome should be quite promising in future.

Moderate English language editing is required. For example, (line 82-83) – instead of …assessments… would be better  …clinical indications..

The manuscript needs moderate language editing by the specialist in the field

Author Response

We thank the reviewer for appreciating our work. We respond to all points one by one. Corrections in the text pertaining to reviewers’ comments are marked in yellow.

  1. In Abstract: The aim of this study should be clearly formulated (differences between HRA and controls, or correlations between microbiota diversity and production of volatile compounds) in the Abstract and Introduction.

We now add the main aims of our study in the lines 18-20 of the abstract. These are further explained in the lines 65-71 of the introduction.

  1. Line 20-21:…. samples from 117 participants undergoing routine colonoscopy…One should specify the time of fecal sampling …≥15 days after conventional colonoscopy (see Materials, line 82)… 

We now add this information in the line 21 of the abstract.

  1. In Results:Firmicutes seem to substantially prevail in normal subjects over HRA cases, when comparing Fig.1. and Fig.2. Is this difference significant?

To explain we add the following statement in the lines 152-154: “Krona plots display the percentiles of classified sequence reads found in all samples of each group and thus comparisons between the NA and HRA groups are – unlike those in all of the following results sections – not subject to statistical evaluation.”

  1. The authors have shown that Proteobacteria,another dominant phylum, are less abundant in normal persons (as shown in Table 2). To justify and characterize the intergroup differences, expecially, broad individual biodiversity of microbiota, one should show in Table 2 minimal and maximal values as well as median percentages for the bacteria of interest.

Table 2 has now been enriched with values indicating the median and the distribution of relative abundance values for each of the bacteria listed. We find that the interquartile range (IQR) (Q1-Q3) and median values are a more suitable indicator of dispersion than minimum/maximum in 16S data, because of the inherent variability of this type of abundance data and the possibility of zero data points (non-detection of bacteria) in some of the samples.

  1. In Table 2, one should also mention the percentage of unclassified NGS reads when analyzing phyla, genera and species. These information losses are usually quite significant.

We now add the following information in the footnote of Table 2: “Percentiles of reads classified at each taxonomic level (mean, SEM): Phylum (mean 99.68%, 0.00014), Family (98.21%, 0.11), Genus (94.91%, 0.21), Species (80.76%, 0.578).”

  1. The numbers of HRA patients tested for volatile compounds (line 124: 18 normal versus 10 HRA subjects) seem to be rather small for reliable correlation analysis between highly variable microbiota diversity (especially, at species levels) and the spectrum of volatile compounds. Therefore, in Table 3, one should add numerical values of the correlation quotients to make the strength of correlations more transparent.

We thank the reviewer for taking note of this. To clarify we revised the titles of Figures 5-8 and Tables 3-4, and the material and methods section lines 139-144. Moreover, reference to “correlation” was substituted with reference to “association” throughout the manuscript to describe VOC-bacteria taxa interactions, which in all cases require distinguishing NA from HRA groups. This analysis is done via binomial logistic regression and does not require the estimation of correlation quotients. For further facilitate understanding all associations shown in Tables 3 and 4 are now displayed in Figures 6 and 8.

  1. In Conclusion, one should emphasize a preliminary character of this study, due to limited number of cases in correlation analysis. Meanwhile, the general idea of correlating intestinal microbiome and metabolome should be quite promising in future.

     We now add the conclusion section 5, in which we provide the suggested information.

  1. Moderate English language editing is required. For example, (line 82-83) – instead of …assessments… would be better …clinical indications..

     We corrected the suggested term (line 79) and went through the material and methods and the other sections to improve the language.

Reviewer 4 Report

This manuscript investigates changes in fecal bacterial species and volatile organic compounds (VOCs) in healthy (non-adenomatous, NA) and CRC-susceptible (high-risk adenomatous, HRA) individuals, used to Establishment to predict colorectal cancer (CRC) risk.  The research topic is in line with the development trend, and the writing is fluent. In particular, this manuscript analyzes the relationship between NA/HRA samples in fecal bacteria and volatile metabolites in detail, and the data analysis and discussion content obtained are of academic reference value. Well-suited for publication after partial correction.

Minor suggestions:

1. Keywords should take precedence over words that appear in the abstract. "pathobionts; nutrients" in the keywords of this manuscript does not even appear in this paper, so it is not suitable as a keyword.

2. This sentence "Higher abundance is indicated with number in bold and the corresponding p-value. SCFAs, short chain fatty acids." is suggested to be moved to the footnote.

3. This sentence "A-E indicate 5 Spearman’s correlations also passing binomial logistic regression analysis (see Figure 6)." is suggested to be moved to the footnote.

4. The "5. Conclusion" paragraph is missing.

5. Suggested reference https://doi.org/10.1016/j.cca.2023.117273, DOI: 10.1097/MD.0000000000020937.

Author Response

The thank the reviewer for appreciating our work.

Regarding the minor suggestions 1-5, below please find our point by point response (manuscript amendments in response to all reviewers are marked in yellow in the manuscript):

  1. Keywords are now amended in the online form (they do not appear in the manuscript) to exclude terms that do not appear in the manuscript.
  2. The pertinent information of Table 2 is now moved to the footnote, as suggested by the reviewer.
  3. The pertinent information of Table 3 is now deleted, instead of being moved to the footnote. To improve clarity though, figure titles for figures 5-8 and Tables 3,4 are amended.
  4. We added an one paragraph "5. Conclusion" section, as suggested by the reviewer.
  5. We added reference 65, as suggested by the reviewer.

Round 2

Reviewer 3 Report

The previous version of this article lacked some data required for strict evaluation of conclusions. Recent corrections have made the end results more convincing and justified. E.g., the issues with small number of cases in experimental group are clarified, appropriate results added.

The article may be published in present form

In current version, the language quality seems to be good.